# Thinkless: LLM Learns When to Think

**Gongfan Fang    Xinyin Ma    Xinchao Wang**[*]
National University of Singapore
gongfan@u.nus.edu, maxinyin@u.nus.edu, xinchao@nus.edu.sg

## Abstract

Reasoning Language Models, capable of extended chain-of-thought reasoning, have demonstrated remarkable performance on tasks requiring complex logical inference. However, applying elaborate reasoning for all queries often results in substantial computational inefficiencies, particularly when many problems admit straightforward solutions. This motivates an open question: Can LLMs learn when to think? To answer this, we propose Thinkless, a learnable framework that empowers an LLM to adaptively select between short-form and long-form reasoning, based on both task complexity and the model's ability. Thinkless is trained under a reinforcement learning paradigm and employs two control tokens, *<short>* for concise responses and *<think>* for detailed reasoning. At the core of our method is a Decoupled Group Relative Policy Optimization (DeGRPO) algorithm, which decomposes the learning objective of hybrid reasoning into two components: (1) a control token loss that governs the selection of the reasoning mode, and (2) a response loss that improves the accuracy of the generated answers. This decoupled formulation enables fine-grained control over the contributions of each objective, stabilizing training and effectively preventing the collapse observed in vanilla GRPO. Empirically, on several benchmarks such as Minerva Algebra, MATH-500, and GSM8K, Thinkless is able to reduce the usage of long-chain thinking by 50% - 90%, significantly improving the efficiency of Reasoning Language Models. The code is available at `https://github.com/VainF/Thinkless`

## 1 Introduction

Reasoning Language Models have exhibited notable effectiveness in solving complex tasks that involve multi-hop reasoning and logic-intensive inference. Their capabilities span a range of domains, such as mathematical problem solving [17, 12, 30] and agentic assistants [39, 5]. A primary factor underlying this success is their ability to perform chain-of-thought reasoning [38], wherein intermediate steps are explicitly generated before arriving at a final answer. While this approach is effective for solving challenging problems, applying it uniformly to all queries, regardless of their complexity or the model's capability can be inefficient. In particular, for questions with straightforward solutions, invoking extended reasoning results in redundant token generation, increased memory footprint, and substantially higher computational cost [11, 7].

In response to these inefficiencies, a growing body of research has sought to enhance the inference efficiency of reasoning models [11, 1, 26, 24, 9, 14, 33]. A prominent direction in this space explores *hybrid reasoning* [4, 36, 2], wherein models dynamically switch between reasoning and non-reasoning modes [41, 2]. Despite promising results, a central challenge persists: determining when a model should engage in elaborate reasoning. Many existing approaches address this by incorporating manually designed heuristics, such as fixed computational budgets [1] or prompt-level control signals like "reasoning on/off" [4, 36]. However, these strategies inherently rely on human prior knowledge and may yield suboptimal or inappropriate control decisions. This underscores a fundamental open

---

[*]Corresponding author

39th Conference on Neural Information Processing Systems (NeurIPS 2025).

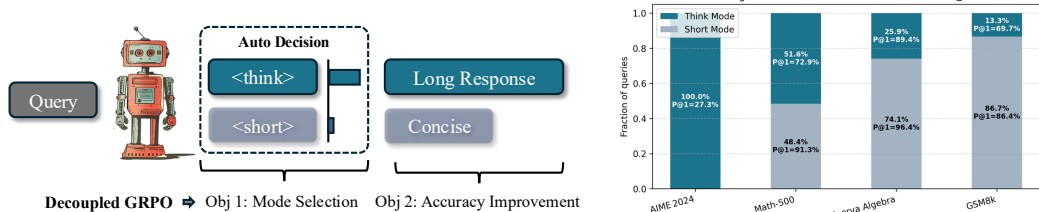

Figure 1: Thinkless learns a hybrid LLM capable of adaptively selecting between thinking and non-thinking inference modes, directed by two special tokens, *<think>* and *<short>*. At the core of our method is a Decoupled Group Relative Policy Optimization, which decomposes and balances the mode selection on the control token and accuracy improvement on the response tokens.

question: Can an LLM *learn* to decide when to think, guided by the complexity of the task and its own capability?

Motivated by this, we explore the fundamental form of hybrid reasoning, where the model is tasked with autonomously deciding whether to generate a short-form or long-form response based on the input query. This decision is guided by three core factors: (1) the complexity of the query, as simpler questions generally merit concise responses, while more intricate ones may necessitate extended reasoning; (2) the capability of the model, since more powerful models are better positioned to employ short reasoning without sacrificing accuracy, whereas less capable models may benefit from longer responses to preserve performance; and (3) the user's tolerance for the trade-off between efficiency and accuracy, which determines the acceptable level of performance degradation when opting for shorter reasoning. Naturally, reinforcement learning [30, 12, 43] offers a framework to unify these factors, as it allows the model to learn from interactions that reflect both environmental feedback and user-defined preferences. Through iterative exploration and reward-driven updates, the model progressively acquires the ability to make autonomous, context-aware decisions about its reasoning strategy, balancing accuracy and efficiency in a dynamic and data-driven manner.

Building on these insights, we propose Thinkless, a reinforcement learning framework designed to train a hybrid reasoning model capable of selecting between short-form and long-form responses. As illustrated in Figure 3, Thinkless employs two control tokens, *<think>* and *<short>*, which are generated as the first token in the model's output to signal the intended inference style. The training comprises two stages: a supervised warm-up phase followed by a reinforcement learning phase.

**Distillation for Warm-up.** In the warm-up phase, the model aligns its response style with the designated control tokens via a distillation process. Specifically, it learns to imitate the behavior of two expert models: a reasoning model and a standard instruction-following model, each conditioned on a specific control token (*<think>* or *<short>*). Additionally, the model is trained on paired long-form and short-form responses for each query, ensuring it can generate both styles with comparable likelihood. This initialization establishes a clear and robust mapping between control tokens and response formats, providing diverse outputs for subsequent reinforcement learning.

**Reinforcement Learning with Decoupled GRPO.** In the reinforcement learning phase, the model is optimized to select the appropriate inference mode based on performance feedback. A natural starting point for this task is the vanilla Group Relative Policy Optimization (GRPO) [12] framework. However, when applied to hybrid reasoning, vanilla GRPO treats all tokens, including the control token and the response tokens uniformly. This introduces a critical imbalance: since the response part often spans hundreds to thousands of tokens and the length of long & short responses varies significantly, the single control token may receive weak and biased gradient signals, ultimately leading to mode collapse at the early stages of training. To this end, we propose a tailored method for hybrid reasoning, termed Decoupled Group Relative Policy Optimization (DeGRPO). As illustrated in Figure 1, DeGRPO explicitly separates the hybrid reasoning objective into two components: (1) Mode Selection, which governs how quickly the policy adapts based on the model's current accuracy; and (2) Accuracy Improvement, which refines the response content to improve answer correctness under the selected reasoning mode. These two components are inherently interdependent, and effective training requires carefully balancing the learning signals for the control token and the response tokens. For instance, if the mode selection policy adapts too aggressively in favor of long-form reasoning, the

model may ignore short-form responses, resulting in insufficient exploration of the potential of short responses. To this end, DeGRPO assigns distinct weights to the control token and response tokens, promoting a more stable and balanced training dynamic. This design not only mitigates mode collapse but also enables the model to learn both accurate outputs and context-sensitive reasoning strategies. After reinforcement learning, the model learns to accurately identify simple queries and respond using the more efficient non-thinking mode. For instance, on benchmarks such as MATH-500, Minerva Algebra and GSM8K dataset, Thinkless reduces the usage of long-form reasoning by 50% - 90%. And on much more challenging tasks like AIME, the model naturally adopts a higher proportion of long-form reasoning.

In conclusion, this work demonstrates that a Reasoning Language Model can learn when to engage in reasoning before generating a response, guided by the proposed Decoupled GRPO method. This adaptive decision-making substantially reduces inference cost while preserving task performance.

## 2 Related Works

**Efficient Reasoning Models.** Reasoning models generate intermediate steps in a chain-of-thought process before producing a final answer [38, 17]. This paradigm offers significant advantages for tasks involving complex computations, logical deduction, and multi-step reasoning. However, excessively long reasoning chains can lead to substantial computational overhead [7, 9, 11]. To mitigate this, recent research has explored strategies to enhance the efficiency of reasoning models without sacrificing accuracy or generalization [26, 24, 1, 18, 13]. Several techniques, such as reinforcement learning with length penalties [1, 24], supervised fine-tuning using variable-length chain-of-thought data [26], and prompt-based methods [7, 40, 3] have been proposed to encourage concise yet effective reasoning paths. Additionally, latent reasoning techniques aim to encode reasoning steps into compact internal representations, thereby reducing token-level computation while maintaining performance [29]. Parallel efforts in knowledge distillation [16, 42, 27, 20, 6] have facilitated the transfer of reasoning capabilities from large to smaller models, while efficient decoding strategies such as predictive decoding and self-consistency optimization have also yielded notable improvements in inference speed and resource utilization [12, 19, 44].

**Hybrid Reasoning.** While prior work has largely focused on compressing reasoning paths to reduce token generation, an alternative path to efficiency is hybrid reasoning, which dynamically adapts the appropriate inference behaviour based on task complexity [2]. This approach allows models to flexibly alternate between short-form responses and long-chain reasoning as needed. Hybrid reasoning can be realized either through collaborative systems involving multiple models [28, 21] or within a single unified model [2, 36, 4]. In multi-model frameworks, routing mechanisms [28] or speculative decoding techniques [21] are commonly employed. For example, a lightweight model may generate a preliminary answer that a larger model verifies or refines. In contrast, unified models are trained to support both reasoning modes and can switch between them via prompt-based control. Some models adopt fixed prompt formats such as "reasoning on/off" to modulate reasoning depth [4]. However, most existing approaches depend on manually crafted heuristics to balance efficiency and performance. In this work, we explore a learning-based alternative, enabling an LLM to automatically determine its inference behavior based on inputs, without relying on manual control.

## 3 Method

The proposed Thinkless is implemented in two stages: (1) **Distillation for Warm-up**, where we fine-tune a pre-trained reasoning model to unify two reasoning styles, and (2) **Reinforcement Learning with DeGRPO**, where the model is trained under a decomposed objective to select the appropriate reasoning mode while improving the response quality.

### 3.1 Distillation for Warm-up

The first step in our framework is to construct a model $\pi_\theta$ capable of generating both short- and long-form responses. We leverage two pre-trained experts for distillation: a reasoning model $\pi_{\text{think}}$, trained to produce detailed chains of thought via step-by-step reasoning; and an instruction-following model $\pi_{\text{short}}$, optimized for generating concise answers aligned with user intent.

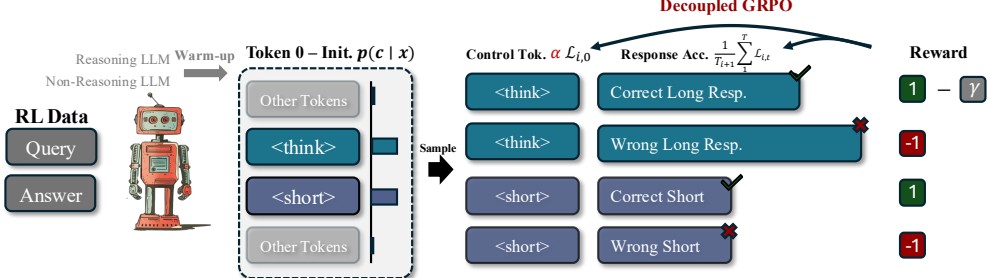

Figure 2: ThinkLess trains a hybrid model that adaptively selects reasoning modes based on task complexity and model capacity. The process begins with distillation, enabling the model to follow control tokens (*<think>* or *<short>*) for guided reasoning. This is followed by reinforcement learning using Decoupled GRPO, which separates training into two objectives: optimizing the control token for effective mode selection and refining the response to improve answer accuracy.

Given a prompt corpus $\mathcal{X} = \{x_i\}_{i=1}^N$, we use these models to generate a synthetic paired dataset:

$$\mathcal{D}_{\text{distill}} = \left\{ \left(x_i, \texttt{<think>}\, a_i^{\text{think}}, \texttt{<short>}\, a_i^{\text{short}}\right) \right\}_{i=1}^N,$$

where $a_i^{\text{think}} = \pi_{\text{long}}(x_i)$ and $a_i^{\text{short}} = \pi_{\text{short}}(x_i)$. Each response is prefixed with a control token $c \in \mathcal{C} = \{\texttt{<short>}, \texttt{<think>}\}$ that conditions the model on the intended reasoning style. We then fine-tune the target reasoning model $\pi_\theta$ on this dataset via supervised fine-tuning (SFT). The objective is to learn a multi-style response distribution conditioned on the control token. This distillation phase ensures that the model is capable of generating both types of responses with high fidelity. Moreover, the paired construction of $\mathcal{D}_{\text{distill}}$ ensures that, the model's response distribution will be balanced. This helps the follow-up RL process to explore different solutions.

## 3.2 Learning When to Think via Decoupled GRPO

After the distillation phase, the model can produce both long- and short-form answers. what it still lacks is a mechanism for *deciding* which reasoning mode suits a particular input $x$. To supply this capability we frame mode selection as a reinforcement-learning problem and optimize a policy $\pi_\theta(c, a \mid x) = \pi_\theta(c \mid x)\,\pi_\theta(a \mid x, c)$, where the first token $c \in \mathcal{C} = \{\texttt{<short>}, \texttt{<think>}\}$ serves as a *control token* that determines the reasoning mode, and the subsequent tokens $(a_{i,1}, \ldots, a_{i,T_i})$ constitute the generated response. For notational convenience, we denote the entire sequence of the $i$-th sample of length $T_i + 1$ as $a_i = (a_{i,0}, \ldots, a_{i,T_i})$, where $a_{i,0} \in \mathcal{C}$ is the control token.

**Reward Design.** Let $y^*$ denote the ground-truth answer corresponding to the input $x$. We consider a minimally designed reward function $r(a, y^*, c)$, which assigns different reward values as follows:

$$r(a, y^*, c) = \begin{cases} 1.0, & \text{if } c = \textit{<short>} \text{ and Extract-Answer}(a) = y^*, \\ 1.0 - \gamma, & \text{if } c = \textit{<think>} \text{ and Extract-Answer}(a) = y^*, \\ -1.0, & \text{if Extract-Answer}(a) \neq y^*, \end{cases}$$

where the $1 > \gamma > 0$ introduces preference to the short correct answer over long responses.

**Decoupled Policy Optimization.** Based on the simple reward function, we adopt a GRPO-based framework [30, 23] for training. Let $\{a_i\}_{i=1}^G$ denote a mini-batch of trajectories sampled from the current policy $\pi_{\theta_{\text{old}}}$. The objective is defined as:

$$\mathcal{J}_{\text{GRPO}}(\theta) = \mathbb{E}_{x, a_i} \left[ \frac{1}{G} \sum_{i=1}^G \left( \frac{1}{T_i+1} \sum_{t=0}^{T_i} \mathcal{L}_{i,t}(\theta) - \beta\, \mathbb{D}_{\text{KL}}\big[\pi_\theta(\cdot \mid x) \,\|\, \pi_{\text{ref}}(\cdot \mid x)\big] \right) \right], \tag{1}$$

where $\mathcal{L}_{i,t}(\theta)$ denotes the token-level surrogate loss formally given by:

$$\mathcal{L}_{i,t}(\theta) = \min \left( \frac{\pi_\theta(a_{i,t} \mid x, a_{i,<t})}{\pi_{\theta_{\text{old}}}(a_{i,t} \mid x, a_{i,<t})} \hat{A}_{i,t},\ \text{clip}\left( \frac{\pi_\theta(a_{i,t} \mid x, a_{i,<t})}{\pi_{\theta_{\text{old}}}(a_{i,t} \mid x, a_{i,<t})}, 1-\epsilon, 1+\epsilon \right) \hat{A}_{i,t} \right) \tag{2}$$

In this work, we compute the relative advantage using $\hat{A}_{i,t} = r - \text{mean}(\mathbf{r})$, following [23]. This choice is motivated by the observation that our training data contains questions of varying difficulty, which can introduce bias when using standard deviation normalization.

When applied to training, the objective in Equation 1 serves two purposes: (1) to learn an appropriate control token for mode selection, and (2) to improve the accuracy of the response tokens:

$$\frac{1}{T_i+1}\sum_{t=0}^{T_i}\mathcal{L}_{i,t}(\theta) = \underbrace{\frac{1}{T_i+1}\mathcal{L}_{i,0}(\theta)}_{\text{Control Token}} + \underbrace{\frac{1}{T_i+1}\sum_{t=1}^{T_i}\mathcal{L}_{i,t}(\theta)}_{\text{Response Tokens}}. \tag{3}$$

For mode selection, the response style is conditioned on the first token $a_{i,0}$, which is trained during the preceding distillation stage. As a result, adjusting the probability of this single control token is sufficient to switch between reasoning modes. Therefore, this token controls the learning of inference mode. For response accuracy, the optimization seeks to improve the generation of the remaining tokens $a_{i,1:T_i}$. However, the above Equation 3 introduces two types of imbalance during optimization: (1) *Mode-Accuracy imbalance* - each trajectory contains only one control token but $T_i$ response tokens, disproportionately reducing the influence of the mode selection compared to the optimization of response accuracy. (2) *Think-Short imbalance* - longer sequences ($T_i^{\text{think}} \gg T_i^{\text{short}}$) further suppress the gradient contribution of the control token due to the normalization factor $1/(T_i+1)$, causing the *<think>* token to be under-optimized compared to the *<short>*. As we will show in the experiment, such imbalance may lead to severe mode-collapse at the beginning of training. To mitigate these imbalances, we propose a decoupled variant of GRPO, denoted as $\mathcal{J}_{\text{DeGRPO}}$, which separately normalizes the contributions of the control and response tokens:

$$\mathcal{J}_{\text{DeGRPO}}(\theta) = \mathbb{E}_{x,a_i}\left[\frac{1}{G}\sum_{i=1}^{G}\left(\underbrace{\alpha\,\mathcal{L}_{i,0}(\theta)}_{\text{Control Token}} + \underbrace{\frac{1}{T_i}\sum_{t=1}^{T_i}\mathcal{L}_{i,t}(\theta)}_{\text{Response Tokens}} - \beta\,\mathbb{D}_{\text{KL}}\left[\pi_\theta(\cdot\mid x)\,\|\,\pi_{\text{ref}}(\cdot\mid x)\right]\right)\right],$$

$$\tag{4}$$

In DeGRPO, the mode selection $\mathcal{L}_{i,0}(\theta)$ and response accuracy improvement $\sum_{t=1}^{T_i}\mathcal{L}_{i,t}(\theta)$ are independently normalized. A length-independent weighting coefficient $\alpha$ is introduced to balance the optimization between mode selection and response generation. This formulation ensures that the control token receives a consistent gradient scale across both short and long sequences, thereby addressing the mode-mode and think-short imbalance, enabling more stable optimization of reasoning-mode selection. As will be shown in the experiments, an appropriately large $\alpha$ can make the mode update more efficient. In our experiment, we set $\alpha = 1/1000$ for stable training.

**Method Summary.** In summary, our method retains the overall structure of the standard GRPO framework. For each query, a mini-batch of samples is drawn from the current policy to estimate the token-level advantages. To address the imbalance between mode selection and response generation, we independently normalize the advantages associated with the control token and response tokens. This separation allows for explicit balancing of their contributions during optimization, leading to more stable and effective training.

## 4 Experiments

### 4.1 Experimental Setups

**LLMs and Datasets.** We employ `DeepSeek-R1-Distill-Qwen-1.5B` as the base model to train a hybrid reasoning policy. To construct long-short paired responses for the distillation phase, we utilize long-form data from open-source datasets [35, 10] generated by the `DeepSeek-R1-671B` model [12], which is well-suited for multi-step reasoning. The corresponding short-form answers are derived using `Qwen2.5-Math-1.5B-Instruct` [41], a compact instruction-tuned model optimized for concise mathematical responses. The hybrid model is direcytly fine-tuned on this paired dataset via supervised fine-tuning, enabling it to accommodate both long and short reasoning styles. The model is then further optimized using the Decoupled Group Relative Policy Optimization (GRPO) algorithm. For the reinforcement learning stage, we primarily use the `DeepScaleR` dataset [25], which comprises approximately 40K labeled examples. For evaluation, we mainly focus on math datasets, including AIME [37], Minerva Algebra [15], MATH-500 [22] and GSM-8K [8].

**Training Details.** All experiments were conducted on a single node with 4 H100 GPUs. For the warmup stage, we set the maximum context length to 16K and clip those overlong samples. We

| Models | Type | AIME 2024 | | Minerva Algebra | | Math-500 | | GSM8K | |
|---|---|---|---|---|---|---|---|---|---|
| | | Pass@1 | #Tokens (Think%) | Pass@1 | #Tokens (Think%) | Pass@1 | #Tokens (Think%) | Pass@1 | #Tokens (Think%) |
| **DeepSeek-R1-1.5B** | | 0.2800 | 18063 | 0.9577 | 3029 | 0.8608 | 5675 | 0.8347 | 1919 |
| **Q-1.5B** | Base LLM | 0.0200 | 1300 | 0.7771 | 933 | 0.5168 | 855 | 0.7022 | 466 |
| **QMath-1.5B** | | 0.1133 | 1128 | 0.9184 | 586 | 0.7604 | 721 | 0.8572 | 447 |
| **Merging-0.5 [34]** | | 0.1333 | 8636 | 0.9292 | 834 | 0.7740 | 1524 | 0.8332 | 601 |
| **Merging-0.6 [34]** | | 0.1733 | 10615 | 0.9321 | 1091 | 0.7900 | 3000 | 0.8381 | 747 |
| **Merging-0.7 [34]** | | 0.1667 | 15854 | 0.9398 | 1834 | 0.8108 | 4347 | 0.8458 | 1201 |
| **CoT-Valve $\alpha = 8$ [26]** | Short CoT | 0.2000 | 10692 | 0.8079 | 1903 | 0.7060 | 3723 | 0.7726 | 773 |
| **CoT-Valve $\alpha = 6$ [26]** | | 0.1933 | 17245 | 0.9468 | 2656 | 0.8024 | 5167 | 0.7970 | 1009 |
| **CoT-Valve $\alpha = 4$ [26]** | | 0.2267 | 17722 | 0.9439 | 2965 | 0.8036 | 5820 | 0.8108 | 1396 |
| **L1$^\dagger$ [1]** | | 0.2267 | 3629 | 0.9452 | 1566 | 0.8144 | 2490 | 0.8431 | 602 |
| **Router Random** | | 0.1467 | 8093 (56.00%) | 0.9211 | 1736 (49.28%) | 0.7608 | 3096 (47.92%) | 0.8205 | 1086 (50.99%) |
| **Router Q-7B** | Hybrid | 0.1667 | 9296 (46.67%) | 0.9250 | 795 (5.64%) | 0.7948 | 2748 (25.00%) | **0.8587** | 563 (2.35%) |
| **Thinkless** | | **0.2733** | 7099 (100.00%) | **0.9459** | 1144 (25.88%) | **0.8184** | 2555 (51.56%) | 0.8418 | 624 (13.31%) |

Table 1: Empirical results of hybrid reasoning. For hybrid algorithms, we additionally report the proportion of queries executed in the thinking mode during evaluation. †: For L1, we control the response length by appending "Think for NUM_TOKENS tokens" to the prompt, where *NUM_TOKENS* was set to match the response length of our model.

train the `DeepSeek-R1-Distill-Qwen-1.5B` for only 1 epoch. The SFT was conducted on the Megatron framework [32]. For the reinforcement learning stage, we extend the context length to 24K. The model was trained only for 600 steps, using the AdamW optimizer with a learning rate of $1 \times 10^{-6}$, $\beta = (0.9, 0.999)$, and a weight decay of 0.01. The batch size is set to 128, with 8 responses sampled for each query, leading to 1024 data points in total. The RL experiments were implemented using the VeRL framework [31]. More details can be found in the Appendix.

## 4.2 Empirical Results on Hybrid Reasoning

> ***Finding 1.*** The learned hybrid reasoning models effectively distinguish complex from simple queries, reducing the use of thinking by 50%–90%.

Table 1 presents a comparison between our method and several existing models or techniques. The first part showcases our baseline model, `DeepSeek-R1-Distill-Qwen-1.5B`, alongside two instruction-following models designed to generate concise answers. It can be observed that the number of tokens generated by reasoning models is typically 5 to 20 times higher than that of standard models. This highlights the potential for significantly improving efficiency by appropriately selecting the reasoning mode. Notably, on challenging datasets such as AIME and MATH-500, reasoning models tend to outperform others by a large margin. However, on simpler datasets like GSM-8K, extended reasoning offers no clear advantage, and the `Qwen2.5-Math-1.5B-Instruct` model with only a 4K context length achieves better results.

The second part of the table illustrates techniques for generating shorter chains of thought. One highly effective approach is model merging [34], where `DeepSeek-R1-Distill-Qwen-1.5B` is interpolated in parameter space with a base model, `Qwen2.5-Math-1.5B`, to obtain a more efficient model without additional training. Another method is the CoT-Valve [26] technique, which applies supervised fine-tuning (SFT) using LoRA. It allows for controllable reasoning length by adjusting the $\alpha$ parameter in LoRA to modulate the magnitude of parameter updates. Both methods provide mechanisms for adjusting reasoning length, such as the interpolation ratio and the LoRA $\alpha$. To show their performance, we sample outputs across a range of lengths. A notable challenge with these methods is that the optimal reasoning length varies significantly across datasets. Consequently, when a good heuristic, such as merging coefficients of 0.6, is determined on one dataset to reduce token usage, it may result in unexpected performance degradation on other benchmarks, such as AIME.

In the final part of our study, we examine hybrid reasoning strategies, focusing on a comparison with router-based approaches. These methods employ a separate large language model (LLM) to assess query difficulty and dispatch inputs to either a reasoning model or a standard model. While effective to some extent, router models are typically independent and lack comprehensive awareness of the target reasoning models, limiting their ability to make confidence-informed decisions. For instance, on the challenging AIME dataset, where even the reasoning model achieves only 28% accuracy, the router struggles to recognize the its difficulty. In contrast, our method jointly considers both input complexity and model capability, dynamically refining the dispatch strategy through direct interaction with real examples. As a result, it achieves efficient and adaptive reasoning without manual tuning.

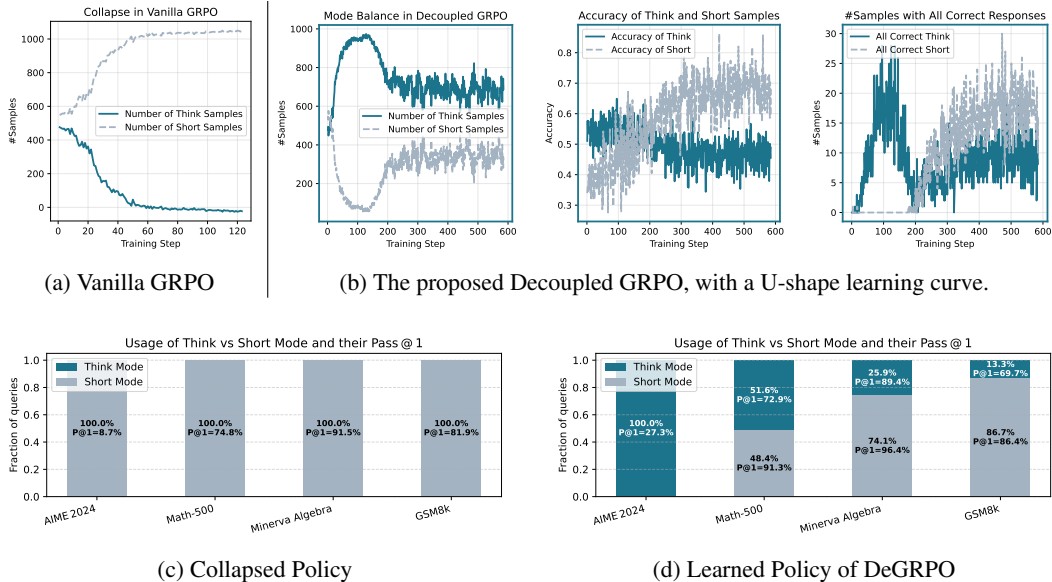

(a) Vanilla GRPO        (b) The proposed Decoupled GRPO, with a U-shape learning curve.

(c) Collapsed Policy        (d) Learned Policy of DeGRPO

Figure 3: Policy–training comparison between vanilla GRPO and decoupled GRPO.

On the Minerva Algebra dataset, our approach activates the reasoning mode for only 25% of the samples, reducing token usage to one-third of the original, while maintaining performance within a 1% margin. In addition, we found that the RL will also compress the length of long responses, since the algorithm will encourage short and correct answers, producing a gradient towards a more compact response.

## 4.3 Training Dynamics in RL

> **Finding 2.** Policy may collapse due to imbalanced update of control tokens in Vanilla GRPO.

**Mode Collapse in RL.** To further analyze how the model learns a reasonable policy, we visualize the training process of RL. Figure 3 (a) illustrates the *Mode Collapse* issue in standard GRPO, where the model develops an excessive preference for either long or short outputs during training. In conventional GRPO, the gradient on the control token is normalized by the total length of the response, which introduces an imbalance between long and short outputs. Specifically, long-chain samples, due to having more tokens, receive slower updates on the `<think>` token, while samples encouraging `<short>` dominate the updates. This imbalance causes the model to collapse rapidly, as shown in Figure 3 (a), the number of generated long-chain responses drops below 10 within just 120 update steps, making it difficult for the model to learn the correct policy. Furthermore, as shown in Figure 3 (c), the model fails to correctly differentiate between samples of varying difficulty, consistently opting for the short-chain reasoning mode.

> **Finding 3.** The U-shape learning curve: The proportion of short-chain samples first drops due to low initial accuracy, then rises after accuracy improvement and mode selection take effect.

**The U-Shape Learning Curve.** To mitigate the collapse issue, we propose a Decoupled GRPO algorithm. Figure 3 (b) illustrates the impact of this decoupling on the training process. We observe a characteristic U-shaped curve in the RL process: the proportion of long-chain outputs initially increases and then gradually decreases. After properly balancing the update weights between the control token and response tokens, the model shows a preference for long-chain reasoning in the early stages of training, primarily because long chains tend to yield higher accuracy. As training progresses, we observe an improvement in the accuracy of short-chain responses. This is driven by two factors: (1) reinforcement learning enhances the generation quality, and (2) the model learns to assign simpler queries to the short-chain reasoning mode. As a result, short-chain responses receive increasingly

| Model | Mode & Teacher | AIME 2024 | | Minerva Algebra | | Math-500 | | GSM8K | |
|---|---|---|---|---|---|---|---|---|---|
| | | Pass@1 | #Tokens | Pass@1 | #Tokens | Pass@1 | #Tokens | Pass@1 | #Tokens |
| **Qwen2.5-1.5B-Instruct** | Short (Base) | 0.0200 | 1300 | 0.7771 | 933 | 0.5168 | 855 | 0.7022 | 466 |
| **Qwen2.5-Math-1.5B-Instruct** | Short (Base) | 0.1133 | 1128 | 0.9184 | 586 | 0.7604 | 721 | 0.8572 | 447 |
| **DeepSeek-R1-1.5B** | Long (Base) | 0.2800 | 18063 | 0.9577 | 3029 | 0.8608 | 5675 | 0.8347 | 1919 |
| **R1-1.5B + OpenR1-97K** | ♦ Long (R1-671B) | 0.2933 | 18880 | 0.9456 | 3239 | 0.8336 | 6780 | **0.8382** | 3423 |
| | ◇ Short (QMath-1.5B) | 0.0733 | 3023 | 0.9001 | 642 | 0.7212 | 1004 | 0.7985 | 433 |
| **R1-1.5B + OpenThoughts-114K** | ♦ Long (R1-671B) | 0.2600 | 17964 | 0.9506 | 3122 | 0.8280 | 6430 | 0.8039 | 2427 |
| | ◇ Short (QMath-1.5B) | 0.0800 | 3807 | 0.8947 | 702 | 0.7136 | 1004 | 0.7886 | 449 |
| **R1-1.5B + OpenThoughts-1M** | ♦ Long (R1-671B) | **0.3067** | 18685 | **0.9530** | 3064 | **0.8360** | 6209 | 0.8253 | 2363 |
| | ◇ Short (QMath-1.5B) | **0.0800** | 3977 | **0.9051** | 690 | **0.7276** | 989 | **0.8202** | 441 |

Table 2: The effectiveness of different SFT datasets during the warm-up stage. Since these models have not yet been optimized via reinforcement learning, the control tokens *<think>* and *<short>* are manually inserted to elicit the desired response patterns.

higher rewards, encouraging the model to further explore the feasibility of short reasoning. This behavior is manifested in the rising proportion of short-chain outputs over time, with many short responses achieving perfect accuracy in the later stages of training. Additionally, we observe a decline in the accuracy of long-chain responses during the latter half of training. This phenomenon is not due to a degradation of the model's reasoning ability but rather a shift in task allocation: more difficult queries, which cannot be solved via short reasoning, are assigned to the long-chain mode, thereby lowering its average accuracy.

> *Finding 4.* The weight $\alpha$ of the control token governs the learning speed of mode selection.

**The Influence of Decoupling.** Building upon the above analysis, we further visualize the effect of decoupling on model behavior. In Figure 4, we show the number of samples that are correctly answered with short responses across all sampled trajectories. We compare the results under two settings: a high control token update weight (0.5) and the weight used in our method (0.001). We observe that with a higher weight on the control token, the all-correct short samples emerge earlier in training. This is because the control token is updated more aggressively, allowing the model to focus more on learning the mode selection. However, excessively fast policy updates can be problematic. For example, some samples may initially yield low accuracy under short responses, but reinforcement learning could eventually improve their short-mode performance. If the model's strategy updates too quickly, it tends to assign such samples prematurely to the long-chain mode, degenerating the algorithm to a simple binary classifier based on initial accuracy rather than a collaborative learning of mode selection and accuracy improvement.

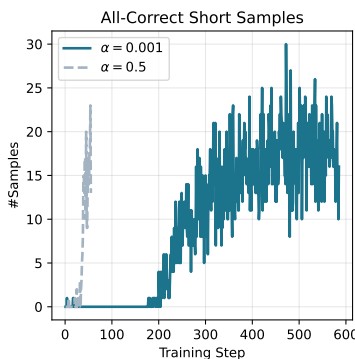

Figure 4: A large token loss coefficient $\alpha$ accelerates the shift in reasoning behavior, leading to the rapid emergence of all-correct short-mode samples.

## 4.4 Details of Warm-up Distillation

> *Finding 5.* Reasoning LLMs are effective short-response learners.

In this work, knowledge distillation is deployed for warm-up, serving as a critical step to equip the model with the basic ability to generate both long chains and short responses. In this section, we provide additional implementation details. Specifically, we consider multiple datasets for distillation. We compare three datasets of increasing scale and domain coverage: (1) OpenR1, a mathematics-only dataset with rigorously verified solutions; (2) OpenThoughts-114K, a compact yet multi-domain dataset labeled by DeepSeek-R1-67B, covering mathematics, science, and programming; and (3) OpenThoughts-1M, a large-scale and diverse collection that subsumes the former two. Table 2 presents the training results on these datasets. Notably, we find that generating short responses using a long-chain reasoning model is relatively straightforward; even with the smallest dataset, OpenR1-97K, the target model successfully learns to produce short outputs. However, this distillation

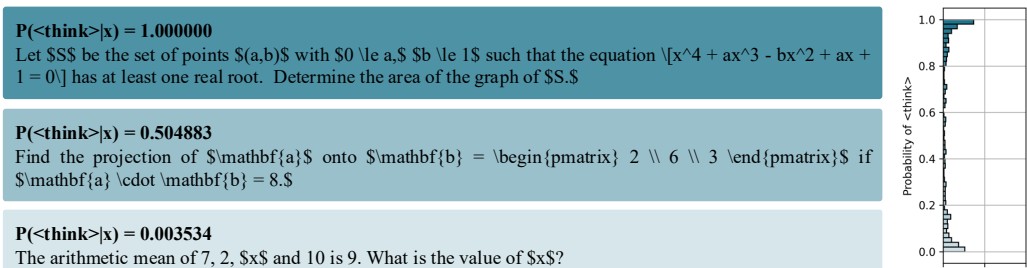

**P(<think>|x) = 1.000000**
Let $S$ be the set of points $(a,b)$ with $0 \le a,$ $b \le 1$ such that the equation $[x^4 + ax^3 - bx^2 + ax + 1 = 0]$ has at least one real root. Determine the area of the graph of $S.$

**P(<think>|x) = 0.504883**
Find the projection of $\mathbf{a}$ onto $\mathbf{b} = \begin{pmatrix} 2 \\ 6 \\ 3 \end{pmatrix}$ if $\mathbf{a} \cdot \mathbf{b} = 8.$

**P(<think>|x) = 0.003534**
The arithmetic mean of 7, 2, $x$ and 10 is 9. What is the value of $x$?

Figure 5: Distribution of the model's probability of emitting *<think>* on MATH-500. The samples with the highest, medium, and lowest probabilities are highlighted. The example with almost 0 thinking score mainly involves straightforward computation, and the query with 1.0 probability relies more on understanding and logical reasoning. More examples and LLM responses can be found in the appendix.

| Model | Easy Problem | Hard Problem |
|---|---|---|
| **R1-1.5B** | 0.9163 (Think) | 0.5977 (Think) |
| **Qwen-1.5B** | 0.7531 (Short) | 0.2835 (Short) |
| **Thinkless-1.5B** | 0.9079 (Short) | 0.6935 (Think) |

(a) Performance on easy and hard problems

| Mode | Length Mean | Length Std | Acc |
|---|---|---|---|
| **Think Mode** | 3185 | 2446 | 0.8436 |
| **Short Mode** | 672 | 465 | 0.7576 |

(b) The output length and accuracy

Table 3: The statistics of think and short modes on the MATH-500 dataset.

process may incur some performance degradation. For instance, on the Math-500 benchmark, the accuracy of the distilled hybrid model is slightly lower than that of the original model, which may affect the downstream performance during the RL phase. For the distillation stage, larger and more comprehensive datasets can lead to improved performance. However, the marginal gains diminish as the dataset size increases. For example, expanding the dataset from 114K to 1M results in only a 1% improvement in long-chain accuracy on the Math-500 benchmark. This work provides a preliminary validation of the effectiveness of simple distillation, and we leave the construction of stronger initial hybrid models as an important direction for future research.

### 4.5 Case Study

> *Finding 6.* The model adaptively adjusts reasoning depth to task difficulty.

**Visualization.** Figure 5 presents a case study on the model's predicted probability of selecting the `<think>` token across the MATH-500 dataset. The distribution reveals that the model makes smooth and hierarchical predictions for queries of different difficulty. In addition, we highlight representative examples corresponding to high, medium, and low confidence levels to illustrate the model's decision behavior. It can be observed that samples assigned to the short reasoning mode are typically simple arithmetic problems that do not require deep or complex reasoning. In contrast, questions routed to the thinking mode tend to be more complex, involving multiple conditions and concepts. Overall, the results reflect a well-calibrated policy that adapts reasoning depth based on task complexity.

**Mode Statistics.** To analyze the learned policy, we categorize the samples in the Math-500 dataset into easy and hard problems based on the decisions made by our model. As shown in Table 3a, we evaluate the performance of the base models (DeepSeek-R1-1.5B [12] and Qwen2.5-1.5B [41]) on these subsets. The model indeed learns to distinguish problem difficulty. At the same time, the RL process explores the potential of short responses across various problems. This results in a significant performance improvement on easy problems, increasing accuracy from 0.7531 to 0.9079. Further, we compared the relative output lengths of the model given the same prompt in Table 3. Across 2,500 trials on Math-500, we found that in 99.8% of cases, the output generated in Think Mode was longer than that in Short Mode.

## 5 Limitations and Future Works

This work presents an effective reinforcement learning framework that enables a hybrid model to adapt its inference mode based on both problem complexity and its own capabilities. However, several limitations remain. For instance, during the warm-up phase, we only validate a simple supervised fine-tuning (SFT) approach without extensive parameter tuning to achieve optimal performance, which results in a slight performance drop in the initial model for reinforcement learning. Exploring better strategies for constructing the hybrid model, such as merging techniques or lightweight fine-tuning methods like LoRA to mitigate catastrophic forgetting, could further enhance the overall performance. In addition, while our algorithm has been validated on DeepScaleR, a dataset containing 40K mathematical problems, future work could expand to a broader range of datasets, incorporating more diverse domains to enable more general and practical hybrid reasoning capabilities.

## 6 Conclusion

This paper proposes a reinforcement learning framework for building a hybrid reasoning model. It autonomously decides whether to generate a short response or engage in long-form reasoning based on the complexity of the input. The core of our approach is a Decoupled GRPO algorithm, which separates the reinforcement learning objective into two components: mode selection on the control token and accuracy improvement on the response tokens. This decoupling enables a more balanced contribution between the two learning objectives. Our method effectively reduces unnecessary long-form reasoning, thereby lowering overall system cost and improving user latency.

## Acknowledgements

This project is supported by the National Research Foundation, Singapore, and Cyber Security Agency of Singapore under its National Cybersecurity R&D Programme and CyberSG R&D Cyber Research Programme Office (Award: CRPO-GC1-NTU-002).

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

# A  Implmentation Details

In this work, we adopt a two-stage training framework: we first fine-tune a reasoning model to generate both long and short responses, and then train it to learn a policy for selecting between the two reasoning modes. Figure 6 shows an example of training a 1.5B LLM. In this section, we provide more details and evaluation.

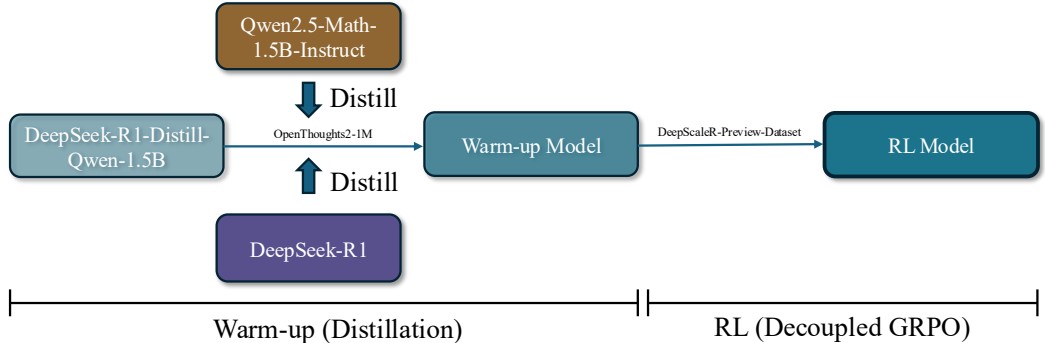

Figure 6: An overview of the SFT and RL training

## A.1  Warmup

We construct an initial Hybrid Reasoning Model through a simple distillation process. The main goal of this process is to establish a connection between control tokens and the model's response style. To achieve this, we train on samples generated by `Qwen2.5-Math-1.5B-Instruct` and `DeepSeek-R1 671B`. Specifically, we use the `OpenThoughts2-1M` dataset, and for each query, we create paired long and short responses from the two teacher models. For long-chain reasoning responses, the DeepSeek model naturally begins with the special token *<think>*, so its outputs can be used directly. For the responses generated by the `Qwen2.5-Math-1.5B-Instruct` model, we prepend an additional *<short>* as the control token.

During training, we include both the long and short training samples of a given query within each batch. We fine-tune the pre-trained model `DeepSeek-R1-Distill-Qwen-1.5B` using a simple supervised fine-tuning (SFT) approach. The training is conducted with the AdamW(lr = 1e-5, weight decay=0.1, betas = (0.9, 0.95)), and we apply a Cosine Annealing Scheduler for learning rate decay, with a minimum learning rate of 1e-6. The model is trained for a single full epoch on a dataset containing 2 million samples, with a batch size of 128, resulting in approximately 17,000 optimization steps. Gradient clipping is applied with a norm threshold of 1.0. The training is conducted on 4 H100 GPUs, and a Distributed Optimizer is used to reduce memory consumption.

## A.2  Reinforcement Learning

In the reinforcement learning stage, we use the hybrid model obtained from the previous step as the foundation. This model is capable of generating both long and short responses with similar probabilities, which facilitates the exploration of various solution strategies across different types of problems. We adopt a simple reward design 1.0, 0.5, -1.0, -1.0, where a reward of 1.0 is assigned to correct short responses, and a relatively lower reward of 0.5 is given to correct long responses. Incorrect responses, regardless of length, receive a reward of -1.0.

We employ the Decoupled GRPO method proposed in this work to learn the policy, which assigns independent, length-agnostic weights to control tokens. We use a simple weight of 0.001 to balance the contributions from control tokens and response tokens. Notably, this setting is equivalent to standard GRPO training with a fixed output length of exactly 1000 tokens. The algorithm is implemented using the VeRL framework. During training, we use a batch size of 128, sampling 8 responses per query, resulting in a total of 1024 training samples per step. Sampling is performed at a temperature of 0.6. The model is trained with a learning rate of 1e-6 for 600 optimization steps.

### A.3 Evaluation

We evaluate our model on four mathematical reasoning benchmarks: AIME [37], Minerva Algebra [15], MATH-500 [22], and GSM8K [8]. The evaluation is conducted using the `lm_eval` framework, with a simple prompting template:

```
Please reason step by step, and put your final answer within
\boxed{}.\n{{problem}}
```

During inference, we adopt a sampling strategy with temperature $= 0.6$ and top_p $= 0.95$. For each problem, we generate 5 completions and report the average accuracy across them. To extract the final answers from LLM responses, we apply a rule-based parsing strategy. We first attempt to extract the value enclosed within `\boxed{}`. If this fails, we fall back to several regular expression patterns to match answers in alternative formats, such as:

```
\*?Final\s+Answer\*?\s*[:=]\s*([0-9.-]+)
```

## B  Social Impact

This work does not pose any significant negative societal impact. We propose an adaptive hybrid model that can reduce long-chain reasoning by 50%–90%, significantly lowering inference costs and power consumption, thereby reducing carbon emissions. Moreover, the adaptive reasoning framework improves the responsiveness of AI systems, particularly in real-time applications. This will help improve the user experience of AI products.

## C  Case Study

We further present additional case studies demonstrating how the learned LLM adopts different inference modes. In Case 1, the input query is a simple arithmetic problem: "1 + 1 = ?". The Thinkless LLM correctly selects the short reasoning mode and responds succinctly. In contrast, the DeepSeek-R1 model, despite its 671B parameters, engages in an unnecessarily lengthy reasoning process involving multiple revisions, resulting in increased computational cost and user latency.

In Case 2, we consider a more complex query that requires step-by-step logical reasoning. In this scenario, both the Thinkless model and DeepSeek-R1 appropriately generate long-chain reasoning to validate various solution paths, producing 1,494 and 1,868 tokens, respectively.

**Case 1: A Simple Question**

**Query:** 1+1=?

---

**Thinkless Response (Short Mode - 99 Tokens)**

<short>
To solve the problem $1 + 1$, we will follow a step-by-step approach:

1. **Identify the numbers involved**: We have the numbers 1 and 1.
2. **Understand the operation**: The operation we need to perform is addition.
3. **Perform the addition**: When we add 1 and 1 together, we get 2.

So, the final answer is $\boxed{2}$.

---

**DeepSeek R1 671B (Think Mode - 376 Tokens)**

Okay, so I need to figure out what 1 plus 1 equals. Let me start by thinking about basic addition. When you add two numbers together, you're combining their values. So if I have one apple and someone gives me another apple, how many apples do I have in total? That should be two apples.

Wait, let me make sure I'm not missing something here. In the simplest terms, addition is taking two quantities and putting them together. So 1 plus 1 means combining a single unit with another single unit. If I use my fingers, holding up one finger on my left hand and one on my right, then when I put them together, I have two fingers up. That seems right.

But maybe there's a trick to this question. Sometimes problems that look easy might have a different answer depending on the context. For example, in binary, 1 + 1 equals 10. But the question doesn't specify a number system, so I should assume it's in the decimal system, which is the default. So in decimal, 1 + 1 is definitely 2.

Another way to check is by using the number line. Starting at 1, if I move one unit to the right, I land on 2. That confirms it. Or using basic arithmetic rules: the sum of 1 and 1 is the next number in the sequence after 1, which is 2.

I guess there's no ambiguity here. Unless the question is trying to be philosophical or something, but mathematically, it's straightforward. So the answer should be 2.
The sum of 1 and 1 is calculated as follows:

$1 + 1 = 2$

In decimal arithmetic, adding the numbers 1 and 1 results in the number 2.

<think> Okay, so I have this simple math problem here: 1 + 1 = ? Hmm, that seems straightforward, but I guess I should think about it carefully to make sure I'm not missing anything. Let me try to visualize it.

First, I know that 1 is just one unit or object. So if I have one apple and someone gives me another apple, how many apples do I have now? Well, I should count them: one, two. So that would make it 2 apples. So, 1 + 1 equals 2. That seems right. But wait, is there another way to think about this? Maybe using objects or fingers? Let me try using my fingers. If I hold up one finger and then another finger, how many fingers am I up on? One, two. Yep, that's 2 again. So that's consistent with my earlier thought. Alternatively, maybe I can use my knowledge of numerals. I know that in the decimal system, each numeral represents a certain value. The numeral 1 represents one, and when you add another 1, you're just moving one more step in the counting sequence. So 1 comes after 0, and then 1 again, and so on. So, 1 plus 1 lands me at 2. Wait, let me think about this in terms of quantities. If I have a set of objects, say two apples, and someone removes one apple, how many are left? That would be 1. So, if I had 1 apple and someone gave me another, I'd have 2. That seems to confirm it. Is there a different perspective? Maybe using number lines. If I start at 1 and move one unit to the right on the number line, I end up at 2. So, that's another way to visualize addition. 1 + 1 = 2 because moving from 1 to 2 is adding one more. I also remember in basic arithmetic, addition is commutative, meaning the order doesn't matter. So 1 + 1 is the same as 1 + 1, which is 2. If I switch them around, 1 + 1 is still 2, so that doesn't change anything. What about if I use different representations? Like, in Roman numerals, 1 is I, and 2 is II. So, I + I would be II, which is 2. That's another way to confirm it. Wait, maybe I can think about it in terms of equations. If I have an equation: 1 + 1 = x. To solve for x, I can subtract 1 from both sides, so x = 1 + 1 - 1, which simplifies to x = 1. But that doesn't make sense because 1 + 1 is 2, not 1. Maybe I made a mistake in the algebra. Let me try again. If I have 1 + 1 = x, then subtracting 1 from both sides would give me x = 1. That can't be right because 1 + 1 is definitely 2. So maybe I can't solve it that way. Maybe I need to use a different approach. Alternatively, maybe I can use the concept of sets. If I have one set with one object and another set with one object, combining them gives me two objects. So, the union of two single-element sets has two elements. Therefore, 1 + 1 equals 2. I guess another way to think about it is through counting. If I count one thing and then another, I have two in total. So, 1 (first) + 1 (second) = 2 (total). That makes sense. Is there a real-world example that could help solidify this? Sure. If I have a dollar and I add another dollar, I have 2. So, 1 dollar plus 1 dollar equals 2 dollars. That's a practical application of addition. What if I use money again? If I have one quarter and I add another quarter, I have 50 cents plus 50 cents, which is 100 cents, or a dollar. So, 1 + 1 = 2 in terms of value. That works too. Maybe I can also think about it in terms of shapes or objects. If I have one square and another square, how many squares do I have? Two squares. So, 1 + 1 = 2. That's another visual confirmation. I guess I'm overcomplicating it, but better safe than sorry. Let me try to recall if I've heard this problem before. I think it's a basic arithmetic problem that's been used in various educational settings. It's often used to introduce the concept of addition for young learners. In summary, through multiple perspectives—counting, using number lines, Roman numerals, set theory, real-world examples, and even different mathematical concepts like commutativity—it consistently points to 1 + 1 = 2. I can't see any other possible answer because the fundamental operations of addition and counting lead to that result.

So, after thinking it through in so many different ways, I feel confident that 1 + 1 equals 2.

**Final Answer** The result of 1 + 1 is $\boxed{2}$ . </think>

To solve the problem (1 + 1), we can approach it from various perspectives:

1. **Counting Objects**: If we have one object and add another, we count them: one, two. Thus, (1 + 1 = 2).

2. **Number Lines**: Starting at 1 and moving one unit to the right on the number line lands us at 2. Hence, (1 + 1 = 2).

3. **Roman Numerals**: In Roman numerals, I (1) + I (1) equals II (2).

4. **Set Theory**: Combining two single-element sets results in a set with two elements, so (1 + 1 = 2).

5. **Real-World Examples**: Adding two dollars gives a total of two dollars, demonstrating (1 + 1 = 2).

6. **Commutative Property**: Since addition is commutative, (1 + 1 = 1 + 1 = 2).

Through multiple perspectives, the result is consistently 2. Final Answer The result of 1 + 1 is $\boxed{2}$ .

**Case 2: A More Complicated Question**

**Query:** What is the smallest positive perfect cube that can be written as the sum of three consecutive integers?

---

**Thinkless Response (Think Mode - 1494 Tokens)**

<think>
Okay, let's try to figure out the smallest positive perfect cube that can be written as the sum of three consecutive integers. Hmm, let's start by understanding the problem step by step.

First, a perfect cube is a number raised to the power of 3. So, we need to find the smallest positive integer $n$ such that $n = a + (a + 1) + (a + 2)$, where $a$ is an integer. Then, $n$ should also be a perfect cube. Let me write that down:

$n = a + (a + 1) + (a + 2)$

Simplifying the right-hand side:

$n = 3a + 3$

So, $n = 3(a + 1)$

Therefore, $n$ must be three times some integer. But $n$ also has to be a perfect cube. So, we need to find the smallest perfect cube that is divisible by 3.

...

I think that's thorough enough. I checked the smaller perfect cubes and confirmed that they don't work, and then verified that 27 works. So, the answer should be 27.

**Final Answer**
The smallest positive perfect cube that can be written as the sum of three consecutive integers is $\boxed{27}$.
</think>

To find the smallest positive perfect cube that can be written as the sum of three consecutive integers, we start by expressing the sum of three consecutive integers. Let the three consecutive integers be $a$, $a + 1$, and $a + 2$. Their sum is:

$$a + (a + 1) + (a + 2) = 3a + 3$$

This sum must be a perfect cube. Let $n$ be the perfect cube, so we have:

$$n = 3(a + 1)$$

Thus, $n$ must be a multiple of 3. The smallest perfect cubes are 1, 8, 27, 64, 125, etc. We check each of these to see if they can be written as $3(a + 1)$:

- For $n = 1$: $3(a + 1) = 1$ gives $a = -\frac{2}{3}$, not an integer.
- For $n = 8$: $3(a + 1) = 8$ gives $a = \frac{5}{3}$, not an integer.
- For $n = 27$: $3(a + 1) = 27$ gives $a = 8$, which is an integer. The three consecutive integers are 8, 9, and 10, and their sum is $8 + 9 + 10 = 27$.

Checking the next perfect cubes (64, 125, etc.) confirms they do not yield integer values for $a$. Therefore, the smallest positive perfect cube that can be written as the sum of three consecutive integers is:

$$\boxed{27}$$

Okay, so I need to find the smallest positive perfect cube that can be written as the sum of three consecutive integers. Let me start by breaking down the problem.

First, let's recall that a perfect cube is a number like 1, 8, 27, 64, etc., which are cubes of integers. The question is asking for the smallest such cube that can be expressed as the sum of three consecutive numbers.

Let me represent three consecutive integers algebraically. If I let the middle integer be $n$, then the three consecutive numbers would be $n1$, $n$, and $n + 1$. Adding these together gives:

$$(n1) + n + (n + 1) = 3n$$

So the sum of three consecutive integers is three times the middle integer. That simplifies things a bit. Therefore, the problem reduces to finding the smallest positive perfect cube that is a multiple of 3, or in other words, the cube must be divisible by 3.

...

Alternatively, if the three consecutive integers are all negative, their sum would be negative, which isn't a positive cube. Therefore, the three integers must be positive. Therefore, 27 is indeed the answer.
Hence, after all this reasoning, I think 27 is the correct answer.
**Final Answer**
The smallest positive perfect cube that can be written as the sum of three consecutive integers is $\boxed{27}$.

...

Thus, 27 is indeed the sum of three consecutive integers and is a perfect cube. Smaller cubes like 1 and 8 are not divisible by 3. Therefore, the smallest positive perfect cube that can be written as the sum of three consecutive integers is:

$$\boxed{27}$$

<think> Okay, so I need to find the smallest positive perfect cube that can be written as the sum of three consecutive integers. Hmm, let's break this down step by step.

First, let me recall what a perfect cube is. A perfect cube is a number that can be expressed as some integer raised to the power of three. So, numbers like 1 ($1^3$), 8 ($2^3$), 27 ($3^3$), 64 ($4^3$), and so on.

Next, the problem says it's the sum of three consecutive integers. Let me denote these three consecutive integers as $n - 1$, $n$, and $n + 1$. So, their sum would be $(n - 1) + n + (n + 1)$. Let me compute that:

$(n - 1) + n + (n + 1) = n - 1 + n + n + 1$

Simplifying that, the -1 and +1 cancel out:

$n + n + n = 3n$

So, the sum of three consecutive integers is just three times the middle number, which is $3n$.

Now, we need this sum, which is $3n$, to be a perfect cube. So, $3n$ must equal some integer cubed. Let me write that as an equation:

$3n = k^3$, where $k$ is a positive integer.

So, $n = \frac{k^3}{3}$. For $n$ to be an integer, $k^3$ must be divisible by 3. Since 3 is a prime number, that means $k$ itself must be divisible by 3. Let me denote $k = 3m$, where $m$ is a positive integer. Substituting back into the equation:

$n = \frac{(3m)^3}{3} = \frac{27m^3}{3} = 9m^3$

So, $n = 9m^3$. Now, the three consecutive integers would be $n - 1 = 9m^3 - 1$, $n = 9m^3$, and $n + 1 = 9m^3 + 1$. Their sum is $3n = 3 \times 9m^3 = 27m^3$, which is indeed a perfect cube.

Now, the question is asking for the smallest positive perfect cube that can be expressed in this way. That means I need the smallest $m$ such that $27m^3$ is positive. Since $m$ is a positive integer, the smallest $m$ is 1.

Plugging $m = 1$ into $27m^3$:

$27 \times 1^3 = 27$

So, 27 is a perfect cube and can be written as the sum of three consecutive integers. Let me verify that:

If $m = 1$, then $n = 9 \times 1^3 = 9$. So, the three consecutive integers are 8, 9, and 10.

Adding them up: $8 + 9 + 10 = 27$. Yep, that works.

But wait, is there a smaller cube than 27 that can be written as the sum of three consecutive integers? Let me check the next smaller cubes.

The cubes below 27 are 1, 8, and 27. 1 is too small because the sum of three consecutive integers is at least 1 + 2 + 3 = 6. 8 is also too small because the next consecutive integers after 1, 2, 3 are 2, 3, 4 which sum to 9, and so on. So, 8 is $2^3$, but as I just thought, it's not achievable because the minimal sum is 6, which is less than 8.

Wait, hold on. Let me think about that again. If $3n = 8$, then $n = 8/3$, which is not an integer. So, 8 cannot be expressed as the sum of three consecutive integers because $n$ must be an integer. Therefore, 8 is out.

Similarly, 1 is too small, as we saw.

So, the next cube is 27. But just to make sure, is 27 the smallest possible?

Wait, another approach: Maybe the sum of three consecutive integers doesn't have to be the same as the cube. Let me think. The sum of three consecutive integers is $3n$, so 27 is the cube, but is there a smaller cube that can be written as $3n$ where $n$ is an integer?

Looking back, 3n has to be a cube. So, 3n = k³, so n = k³ / 3. So, n must be an integer, which requires k³ divisible by 3. So, k must be divisible by 3, so k = 3m, so n = 9m³. Thus, the smallest k is 3, giving 27. So, yes, 27 is the smallest.

Alternatively, perhaps I can think of it differently. Let me set up the equation:

Sum of three consecutive integers: $(n - 1) + n + (n + 1) = 3n$. So, 3n is a cube. Therefore, 3 divides the cube, so 3 divides n as well. So, n must be a multiple of 3, meaning n = 3k, so 3n = 9k. Therefore, 9k must be a cube. So, 9k is a cube, which means k must be a multiple of 3, so k = 3m, so 9k = 27m, which is a cube. Therefore, the minimal m is 1, so 27 is the minimal such cube.

So, yes, 27 is the smallest positive perfect cube that can be written as the sum of three consecutive integers.

Wait a second, let me just confirm with another approach. Let me consider the sum of three consecutive integers as $n - 1 + n + n + 1 = 3n$. So, 3n must be a cube. So, 3n = k³, so n = k³ / 3. So, n must be integer, so k³ must be divisible by 3. Therefore, k must be a multiple of 3, so k = 3m, so n = (27m³)/3 = 9m³. So, n must be 9m³, which is integer for any integer m. So, the sum is 3n = 27m³, which is a cube. So, the smallest cube is when m=1, which gives 27. So, yes, 27 is the minimal.

Alternatively, maybe I can list some cubes and see if any smaller cube can be written as 3n.

Cubes: 1, 8, 27, 64, 125, etc.

Check if any of these can be written as 3n:

1: 3n=1 => n=1/3, not integer.

8: 3n=8 => n=8/3, not integer.

27: 3n=27 => n=9, which is integer.

64: 3n=64 => n=64/321.333, not integer.

125: 3n=125 => n41.666, not integer.

So, indeed, 27 is the first cube where n is an integer. Therefore, 27 is the minimal.

So, the answer is 27.

**Final Answer** The smallest positive perfect cube that can be written as the sum of three consecutive integers is $\boxed{27}$. </think>

To find the smallest positive perfect cube that can be written as the sum of three consecutive integers, we start by denoting these three consecutive integers as $n - 1$, $n$, and $n + 1$. Their sum is:

$$(n - 1) + n + (n + 1) = 3n$$

We need this sum, which is $3n$, to be a perfect cube. Let $3n = k^3$ where $k$ is a positive integer. This implies:

$$n = \frac{k^3}{3}$$

For $n$ to be an integer, $k^3$ must be divisible by 3. Since 3 is a prime number, $k$ must be divisible by 3. Let $k = 3m$ where $m$ is a positive integer. Substituting back, we get:

$$n = \frac{(3m)^3}{3} = 9m^3$$

Thus, the three consecutive integers are $9m^3 - 1$, $9m^3$, and $9m^3 + 1$. Their sum is:

$$3 \times 9m^3 = 27m^3$$

We need the smallest positive perfect cube. The smallest $m$ is 1, giving:

$$27 \times 1^3 = 27$$

Verifying, the three consecutive integers are 8, 9, and 10, which sum to 27. Therefore, the smallest positive perfect cube that can be written as the sum of three consecutive integers is:

$$\boxed{27}$$

