# OpenReview forum: "Thinkless: LLM Learns When to Think"
_NeurIPS.cc/2025/Conference — NeurIPS 2025 poster_

### Official Review · Reviewer_q6mt · 2025-06-24

**Clarity:** 3
**Significance:** 2
**Originality:** 3
**Rating:** 4
**Confidence:** 4

**Summary:**

The paper introduces a framework designed to teach LLMs to decide when to use short think versus long think. The model learns to select between a <think> token for long reasoning and a <short> token for concise responses. The work also proposes a reinforcement learning algorithm, Decoupled GRPO (DeGRPO), which solves the critical issue of "mode collapse" by separately optimizing the policy for mode selection and answer accuracy.

**Questions:**

1. When observing the collapse in Vanilla GRPO, what is the final Pass@1 accuracy of the collapsed policy?
2. Reward shaping: have you tried finegrained reward shaping to control the response length?
3. Could you do log-scale sweep of alpha values (e.g., 0.1, 0.01, 0.001, 0.0001)?

**Ethical Concerns:**

["NO or VERY MINOR ethics concerns only"]

**Final Justification:**

This paper addresses an important problem and makes a meaningful contribution.
My main concerns are:
(1) The proposed method appears to be a relatively straightforward extension of existing approaches, with only minor modifications, lacking originality.
(2) To the best of my knowledge, the reported performance is somewhat lower than that of certain concurrent works (the author trained for longer steps to mitigate this in the rebuttal).
That said, I still consider the paper to be above the borderline acceptance threshold and would maintain my current score.

**Limitations:**

Yes

**Quality:**

3

**Strengths And Weaknesses:**

Strengths
1. The algorithm is well-motivated. The authors show that GRPO fails due to mode collapse and correctly identify the token imbalance as the cause. The proposed DeGRPO is a direct solution. It decouples the loss for the control token from the response tokens, which makes the algorithm a targeted fix for a clearly documented empirical problem.
2. The experiments are well-designed. Author provides experiments covering many areas.
3. This paper provides many insightful findings.

Weaknesses
1.  Performance Degradation: The main weakness is that the method's efficiency gains come at the cost of performance. On AIME 2024, the final Thinkless model achieves a Pass@1 of 0.2733, which is lower than the 0.2800 of its own base model (DeepSeek-R1-1.5B). While token usage is reduced, this negative trade-off is concerning, especially as other recent methods for reasoning truncation have demonstrated the ability to simultaneously reduce tokens and improve accuracy. For instance, concurrent work such as Laser-D and Laser-DE achieve a 6.1% improvement on AIME2024 while reducing token usage by 63%. While the methods differ, this comparison suggests the proposed approach may be suboptimal.
2. Limited Scope of Evaluation: The experiments are limited to math reasoning. This leaves the method's generalizability to other domains like commonsense reasoning or code generation.
3. Dependence on Distillation Quality: The performance of the RL stage appears dependent on the quality of the initial "warm-up" model from the distillation phase. The paper does not adequately explore this sensitivity.
4. The analysis of key hyperparameters is not thorough enough. For example, when examining the control token update weight, the paper only compares 0.5 and 0.001. This is a large gap, and a more progressive sweep is needed to understand the trend and sensitivity of this critical parameter.

---

> ### Author Rebuttal · Authors · 2025-07-30
>
> We sincerely appreciate Reviewer q6mt's comments and feedback regarding performance, evaluation scope, warm-up quality, and hyperparameter tuning. We will carefully revise the manuscript in accordance with the reviewer’s suggestions and include the additional results presented below.
>
> ---
>
> > **W1:** *Performance Degradation: The main weakness is that the method's efficiency gains come at the cost of performance. On AIME 2024, the final Thinkless model achieves a Pass@1 of 0.2733, which is lower than the 0.2800 of its own base model (DeepSeek-R1-1.5B). While token usage is reduced, this negative trade-off is concerning, especially as other recent methods for reasoning truncation have demonstrated the ability to simultaneously reduce tokens and improve accuracy. For instance, concurrent work such as Laser-D and Laser-DE achieve a 6.1% improvement on AIME2024 while reducing token usage by 63%. While the methods differ, this comparison suggests the proposed approach may be suboptimal.*
>
> **Response to W1:**  Thank you for the comment. The accuracy can be improved by increasing the number of training steps. Our method typically requires more training iterations, as the algorithm initially prioritizes learning the mode selection strategy (i.e., forming the U-shaped curve in the early 300 steps), followed by accuracy improvement under the learned policy in later stages. To further investigate the issue raised by the reviewer, we extended training by an additional 200 steps. As shown, the model continues to improve its performance on long chains during this later phase. However, the improvement may be owing to RL itself rather than our method. We will discuss this in the revised version.
>
> | Training Steps | AIME | Minerva Algebra | Math-500 | GSM8K |
> | :-- | :--: | :--: | :--: | :--: |
> | 600 Steps | 0.2733 | 0.9459 | 0.8184 | 0.8418 |
> | + 200 Steps | 0.2866 | 0.9498 | 0.8292 | 0.8502 |
>
> ---
>
> > **W2:** *Limited Scope of Evaluation: The experiments are limited to math reasoning. This leaves the method's generalizability to other domains like commonsense reasoning or code generation.*
>
> **Response to W2:** Thank you for the comment. In response to the suggestion, we have conducted additional evaluations on CommonsenseQA to assess both the original DeepSeek-R1-Distill-Qwen-1.5B and our RL model. Our method achieves a 36% reduction of thinking on this commonsense reasoning task. However, we acknowledge that this result does not imply full generalizability across all tasks as mentioned in our limitation section, as the RL dataset used in our work comprises only 40K math problems. For instance, we observed that the model may struggle on some general tasks such as If-Eval, where only 10% of the samples are considered easy.
>
> | Commonsense QA | DeepSeek-R1-Distill-Qwen-1.5B | Ours |
> | :-- | :--: | :--: |
> | Acc | 0.4426 | 0.4523 |
> | Num Tokens (Think%) | 1538 | 1116 |
> | Percentage of Think | 100.00% | 64.42% |
>
> ---
>
> > **W3:** *Dependence on Distillation Quality: The performance of the RL stage appears dependent on the quality of the initial "warm-up" model from the distillation phase. The paper does not adequately explore this sensitivity.*
>
> **Response to W3:** Thank you for your question. The performance of the teacher models during the warm-up phase indeed influences downstream results, as it determines the initial state for reinforcement learning. In the paper, we explored training data sizes ranging from 100K to 1000K and selected the best model among them. Following your suggestion, we further conducted RL training on different warm-up models. Due to the limited time and computational resources during the rebuttal period, we were only able to train for 400K steps. As shown, our method is still able to learn reasonable policies on this relatively weaker initial models.
>
> | Initial Models | AIME Acc | AIME Len | Algebra Acc | Algebra Len | Math-500 Acc | Math-500 Len | GSM8K Acc | GSM8K Len |
> | :-- | :--: | :--: | :--: | :--: | :--: | :--: | :--: | :--: |
> | OpenThoughts-1M Warmup (Reported in the paper) | **0.2733** | 7099 (100.00%) | **0.9459** |  1144 (25.88%) |  0.8184 | 2555 (51.56%) | **0.8418** |  624 (13.31%) |
> | OpenR1-97K Warmup (400K) | 0.2600 | 7336 (100.00%) | 0.9375 | 1236 (25.59%) | **0.8236** | 2499 (54.44%) | 0.8415 | 618 (13.43%) |
>
>
> ---
>
> > **W4:** *The analysis of key hyperparameters is not thorough enough. For example, when examining the control token update weight, the paper only compares 0.5 and 0.001. This is a large gap, and a more progressive sweep is needed to understand the trend and sensitivity of this critical parameter.*
>
> **Response to W4:** Thank you for the suggestion. We have added additional hyperparameter experiments covering α values of 0.1, 0.01, 0.001, and 0.0001. As figures are not permitted in the rebuttal, we instead present key observations from training in the table below. Here, the inflection point refers to the number of training steps required to reach the lowest or highest value in the U-shaped learning curve, as illustrated in Figure 3b of the main paper. A smaller inflection point indicates faster learning of mode switching. The results show that larger $\alpha$ values lead to an earlier inflection point in the U-shaped training curve, as well as an earlier emergence of all-correct short samples. This suggests that the model places greater emphasis on learning when to switch modes. In contrast, smaller α values result in slower mode switching. We apologize for not being able to complete a full sweep of training experiments due to time constraints during the rebuttal period. We will include full visualizations of the training curves in the revised version.
>
> | Hyperparameter $\alpha$ | 0.1 | 0.01 | 0.001 | 0.0001 |
> | :--  | :--: | :--: | :--: | :--: |
> | Inflection point (Steps)* | 42 | 76 | 132 | 224 |
> | Emergence of All-correct Short Samples (Steps) | 54 | 102 | 200 | 341 |
>
> ---
>
> > **Q1:** *When observing the collapse in Vanilla GRPO, what is the final Pass@1 accuracy of the collapsed policy?*
>
> **Response to Q1:** Thank you for the valuable question. The Pass@1 results are presented in Figure 3c of the main paper. We will revise the figure caption to better highlight these results. For your convenience, we report the corresponding values in the table below.
>
> | Method | AIME | Minerva Algebra | Math-500 | GSM8K |
> | :-- | :--: | :--: | :--: | :--: |
> | Collapsed | 0.0870 | 0.9150 | 0.7480 | 0.8190 |
> | Ours | 0.2733 | 0.9459 | 0.8184 | 0.8418 |
>
> ---
>
> > **Q2:** *Reward shaping: have you tried finegrained reward shaping to control the response length?*
>
> **Response to Q2:** Thank you for the suggestion. We agree that adjusting the reward can help control response length. Due to time constraints, we conducted quick finetuning from the 300-step checkpoint, increasing the reward for long and correct samples to 0.7. The results, shown in the following table, indicate that this adjustment effectively improves accuracy, although it leads to an increase in the number of generated tokens.
>
> | $r_\text{think,correct}$ | AIME | Minerva Algebra | Math-500 | GSM8K |
> | :-- | :--: | :--: | :--: | :--: |
> | 0.5 | 0.2733 / 7099 | 0.9459 / 1144 | 0.8184 / 2555 | 0.8418 / 624 |
> | 0.5 $\rightarrow$ 0.7 | 0.2733 / 9606 | 0.9521 / 1304 | 0.8357 / 3132 | 0.8436 / 744 |
>
> ---
>
> > **Q3:** *Could you do log-scale sweep of alpha values (e.g., 0.1, 0.01, 0.001, 0.0001)?*
>
> **Response to Q3:** Thanks for the suggestion. The results are available in the response to Weakness 4.

---

> > ### Comment · Reviewer_q6mt · 2025-08-04
> >
> > I thank the authors for their extensive response and experiments, which addressed some of my concerns. I encourage them to improve the paper by adding these new experiments.

---

> > > ### Author Response · Authors · 2025-08-04
> > >
> > > We sincerely appreciate the valuable suggestions from Reviewer q6mt. As pointed out by Reviewer q6mt, there is still room for improvement in our work. We will carefully revise the paper and incorporate all the suggested results, including continual training, additional evaluation metrics, and training curves under different hyperparameter settings, following the reviewer's suggestions.
> > >
> > > Best regards,
> > > The Authors of Submission #2048

---

### Official Review · Reviewer_Atrz · 2025-06-30

**Clarity:** 3
**Significance:** 3
**Originality:** 3
**Rating:** 5
**Confidence:** 4

**Summary:**

This paper introduces Thinkless, a reinforcement learning framework that enables Large Language Models to adaptively choose between detailed chain-of-thought reasoning and direct short-form responses based on task complexity. The authors propose a two-stage training approach: first, knowledge distillation using control tokens (<think> for long reasoning and <short> for concise answers) to establish dual response capabilities; second, a novel Decoupled Group Relative Policy Optimization (DeGRPO) algorithm that separates the learning objectives for mode selection and response accuracy, addressing the gradient imbalance issues in vanilla GRPO that lead to mode collapse. Experiments on mathematical datasets (AIME, MATH-500, Minerva Algebra, GSM8K) demonstrate that Thinkless reduces the use of long-form reasoning by 50-90% while maintaining comparable accuracy (within 1% degradation), achieving significant computational savings by intelligently allocating reasoning resources—using extensive thinking only when necessary for complex problems while providing quick responses for simpler queries.

**Questions:**

As I stated in Weaknesses

**Ethical Concerns:**

["NO or VERY MINOR ethics concerns only"]

**Final Justification:**

Thanks for the author's reply, which has solved my problem. I will keep my score.

**Limitations:**

yes

**Quality:**

3

**Strengths And Weaknesses:**

## Strengths
1、 The authors address an interesting problem: how to enable models to autonomously think based on question difficulty, thereby answering more efficiently. This is a highly interesting problem, and the authors propose an effective solution that can contribute to the community.

2、The authors design control tokens to manage two different thinking modes and discover that directly using the original GRPO algorithm leads to mode collapse, where the model only learns the short mode rather than adapting. Based on this, the authors propose Decoupled GRPO and improve both parts of the loss function, achieving excellent experimental results.

## Weaknesses and Questions
1、I am curious about what would happen if we directly proceed with reinforcement learning training without the first-stage warm-up? A related question: after the first-stage warm-up, would long responses still appear under the short mode in the second-stage reinforcement learning training?

2、Similar to the previous question, in the reward function, when the model uses short mode and gets the correct answer, it receives a reward of 1. Could this design lead the model to prefer outputting <short> while still using long responses to answer questions? I would like to see analytical experiments on the changes in average response lengths for both short and long modes.

3、In the second plot of Figure 3-b, it seems that the accuracy of think mode shows a significant decline after 200 steps. Why does this happen?

---

> ### Author Rebuttal · Authors · 2025-07-30
>
> We sincerely appreciate the reviewer’s valuable comments and suggestions. We will revise the manuscript to include a detailed explanation of the necessity of SFT prior to RL, as well as a discussion of the hacking issue.
>
> ---
>
> > **W1:** *1) I am curious about what would happen if we directly proceed with reinforcement learning training without the first-stage warm-up? 2) A related question: after the first-stage warm-up, would long responses still appear under the short mode in the second-stage reinforcement learning training?*
>
> **Response to W1:**
> 1. Thank you for your question. Directly training with reinforcement learning (RL) poses two key challenges: (a) the Short mode is rarely sampled, and (b) the responses generated under the Short mode often fail to reflect the intended brief reply style, and we need additional constraints to encourage short responses. Therefore, we employ supervised fine-tuning (SFT) to explicitly associate each control token with its corresponding behavior. This also allows us to balance the sampling probability between response styles (e.g., 50%-50%), ensuring both modes are sufficiently explored during training.
> 2. As shown in the table below, we compare the response lengths under the same prompt from Math-500. We did observe that in 99.8% of validation samples, the response length in Think Mode was longer than in Short Mode. The remaining 0.2% unexpected cases exhibited endless outputs, which might be caused by the SFT, where not all cases were covered by the training distribution.
>
> | Length Stats | len(Think)>len(Short) |
> |:--| :--: |
> | Warmup Model | 99.7% |
> | RL Model | 99.8% |
>
> ---
>
> > **W2:** *Similar to the previous question, in the reward function, when the model uses short mode and gets the correct answer, it receives a reward of 1. Could this design lead the model to prefer outputting [object Object] while still using long responses to answer questions? I would like to see analytical experiments on the changes in average response lengths for both short and long modes.*
>
> **Response to W2:** Thank you for the question. We agree that such hacking behavior is indeed a potential concern. However, SFT can effectively mitigate this issue. As demonstrated in the statistical results above, the model's behavior after SFT aligns well with our expectations, on the validation set, Think Mode consistently produces longer outputs, while Short Mode yields shorter ones. Additionally, it is also possible to introduce explicit length penalties as further constraints to discourage such behavior. However, in pursuit of a simpler design, we adopt only the most basic SFT approach in this work.
>
> ---
>
> > **W3:** *In the second plot of Figure 3-b, it seems that the accuracy of think mode shows a significant decline after 200 steps. Why does this happen?*
>
> **Response to W3:** Thank you for your question. This is indeed a particularly interesting phenomenon observed in our RL setup. The apparent drop in accuracy for longer chains actually arises because the algorithm tends to assign more challenging problems to the Think mode. As a result, the model shows lower accuracy on these harder examples, leading to a perceived "decline" in accuracy. For example, we report the models’ accuracy on easy and hard problems in Math-500, as categorized according to the decision of our model. We observe that the accuracy on hard problems (0.6935) is indeed lower than that on easy ones (0.9079). This means the RL model indeed learns how to assess the problem difficulty.
>
>
> |  | Acc on Easy Problems   | Acc on Hard Problems |
> |:--| :--: | :--: |
> | R1-1.5B | 0.9163 (Think) | 0.5977 (Think) |
> | Qwen-1.5B | 0.7531 (Short) | 0.2835 (Short) |
> | Ours | 0.9079 (Short) | 0.6935 (Think) |

---

### Official Review · Reviewer_5tcS · 2025-07-02

**Clarity:** 3
**Significance:** 2
**Originality:** 2
**Rating:** 4
**Confidence:** 3

**Summary:**

This paper proposes a two-stage approach to teach large language models (LLMs) when to engage in longer reasoning. In the first stage, supervised fine-tuning on distilled examples introduces two new control tokens and teaches the model their intended semantics. In the second stage, reinforcement learning trains the model to use these tokens effectively for efficient problem solving. The paper also introduces corrections to the reinforcement learning procedure to prevent mode collapse.

**Questions:**

- How does the proposed method compare quantitatively to other relevant baselines? Why are there no reported results for some of the methods cited in the related work section?
- In Figure 3b, there appears to be some degradation in performance on examples using the <think> token as training progresses. What might explain this decline? Should we expect the accuracy on <think> samples (the default mode) to remain relatively stable throughout training, while accuracy on <short> samples to improve over time?

**Ethical Concerns:**

["NO or VERY MINOR ethics concerns only"]

**Limitations:**

yes

**Quality:**

3

**Strengths And Weaknesses:**

Strengths:

- The paper presents a clear and straightforward idea that brings measurable benefits.
- The proposed reinforcement learning formulation, with its modifications to mitigate mode collapse, is sound and well-motivated.
- I appreciate that the experimental design is structured around specific findings, with empirical results provided to support each of them.

Weaknesses:

- The novelty of the work appears limited. There is substantial prior research on controlling when language models should “think” more, such as “Think Before You Speak: Training Language Models with Pause Tokens” and “Training Language Models to Reason Efficiently”. In addition, there are several recent and concurrent works in this space that are not cited as related work.
While the paper cites related baselines ([1, 13, 18, 24]), it does not actually compare to them in the experiments—only [26] is included as a baseline. This makes it hard to judge the empirical significance of the proposed approach relative to existing methods.
- Finding 1 claims that the model can distinguish simple from complex queries, but the evidence for this is indirect. Table 1 only shows pass@1 and reasoning trace lengths. A stronger evaluation would involve explicitly classifying queries into simple and complex (e.g., by baseline success rate) and demonstrating that Thinkless activates the <think> mode significantly more often on harder instances, similar to Fig. 5 but stratified by problem complexity.

---

> ### Author Rebuttal · Authors · 2025-07-30
>
> We sincerely appreciate the valuable feedback from Reviewer 5tcS. We will make our best effort to enhance the novelty of this work and provide a more comprehensive discussion of related works. The following results will also be included in the revised version.
>
> ---
>
> > **W1:** *The novelty of the work appears limited. There is substantial prior research on controlling when language models should “think” more, such as “Think Before You Speak: Training Language Models with Pause Tokens” and “Training Language Models to Reason Efficiently”. In addition, there are several recent and concurrent works in this space that are not cited as related work. While the paper cites related baselines ([1, 13, 18, 24]), it does not actually compare to them in the experiments—only [26] is included as a baseline. This makes it hard to judge the empirical significance of the proposed approach relative to existing methods.*
>
> **Response to W1:** Thank you for your valuable feedback. We also sincerely appreciate the reviewer for pointing out relevant works. We will include all the mentioned citations and provide a more detailed discussion in the revised version. Following the reviewer’s advice, we have added comparisons with additional baselines in our response to Q1 below.
>
> ---
>
> > **Q1:** *How does the proposed method compare quantitatively to other relevant baselines? Why are there no reported results for some of the methods cited in the related work section?*
>
> **Response to Q1:** Thank you for the comment. We indeed encountered challenges when evaluating the baselines. For instance, most prior works are based on 32B models, where RL can be too expensive for us to train the hybrid model. In response to the suggestion, we provide additional results for the length-controllable method L1 (Pranjal Aggarwal & Sean Welleck). Following the official instructions, we control the response length of `L1-Qwen-1.5B-Exact` by appending "Think for NUM_TOKENS tokens." to the prompt, where NUM_TOKENS was set to match the response length of our model.
>
> However, we found that controlling the response length through prompting remains challenging. For example, the prompt "Think for 7099 tokens." resulted in an average output length of only 3,626 tokens on AIME'24. On other tasks, the prompt can successfully produce the desired response length. However, one potential advantage of our method is its ability to automatically determine the reasoning budget without relying on manually specified priors. We will include these results in the paper and provide a more detailed analysis in the revised version.
>
> | Methods | AIME Acc | AIME Len | Algebra Acc | Algebra Len | Math-500 Acc | Math-500 Len | GSM8K Acc | GSM8K Len |
> | :-- | :--: | :--: | :--: | :--: | :--: | :--: | :--: | :--: |
> | L1 (Pranjal Aggarwal & Sean Welleck) | 0.2267 | 3629 |  0.9452 | 1566 | 0.8144 | 2490 | **0.8431** | 602 |
> | Ours (Reported in the paper) | **0.2733** | 7099 (100.00%) | **0.9459** |  1144 (25.88%) |  **0.8184** | 2555 (51.56%) | 0.8418 |  624 (13.31%) |
>
> ---
>
> > **W2:** *Finding 1 claims that the model can distinguish simple from complex queries, but the evidence for this is indirect. Table 1 only shows pass@1 and reasoning trace lengths. A stronger evaluation would involve explicitly classifying queries into simple and complex (e.g., by baseline success rate) and demonstrating that Thinkless activates the [object Object] mode significantly more often on harder instances, similar to Fig. 5 but stratified by problem complexity.*
>
> **Response to W2:**  Thank you very much for the suggestion. To explore this issue, we provide a supplementary experiment: We first divided the dataset into "hard" and "easy" samples based on our model's decision, and then evaluated the performance of the base models (DeepSeek-R1-1.5B and Qwen-Math-1.5B) on each subset. The first observation is that the RL model indeed learns to distinguish problem difficulty, where the accuracy on hard problems is lower than that on easy ones. Moreover, the RL process also exploits the potential of short responses on those easy problems, making the short mode of Thinkless competitive to a reasoning model (0.9079 vs. 0.9163)
>
>
> |  | Easy Problem   | Hard Problem |
> |:--| :--: | :--: |
> | R1-1.5B | 0.9163 (Think) | 0.5977 (Think) |
> | Qwen-1.5B | 0.7531 (Short) | 0.2835 (Short) |
> | Ours | 0.9079 (Short) | 0.6935 (Think) |
>
> ---
>
> > **Q2:** *In Figure 3b, there appears to be some degradation in performance on examples using the [object Object] token as training progresses. What might explain this decline? Should we expect the accuracy on [object Object] samples (the default mode) to remain relatively stable throughout training, while accuracy on [object Object] samples to improve over time?*
>
> **Response to Q2:** Thank you for the comment. We also observed this interesting phenomenon during our experiments and have provided an analysis in the paper (Lines 267-271). The observed performance drop is actually due to the algorithm assigning the most difficult samples to the `<think>` mode, just as shown in response to W2, where the accuracy of the think mode is 0.6935, while the short mode achieves 0.9079. So, this is an expected behaviour, which means the model indeed learns how to assess the difficulty of problems. We will further include this result as a more detailed analysis.

---

> > ### Comment · Reviewer_5tcS · 2025-08-04
> >
> > I thank the authors for their response. Overall, I find the answers to W2 and Q2 convincing, which reassures me that the paper meets the rigor required for acceptance. That said, I would encourage the authors to expand on the quantitative relationship between task difficulty (easy vs. hard), the use of the think token, and model performance, particularly during training. I still find it somewhat unfortunate that more baselines could not be included, especially given the breadth of related work.

---

> > > ### Author Response · Authors · 2025-08-04
> > >
> > > We are sincerely grateful to Reviewer 5tcS for the insightful comments and kind suggestions, which offer valuable guidance for improving our work. We will incorporate all the suggested results to provide a more comprehensive discussion on task difficulty, the control tokens, and model performance. We will also make an effort to explore more baseline methods in the coming days and share the results here if possible.
> > >
> > > Best regards,
> > > The Authors of Submission #2048

---

### Official Review · Reviewer_x1rq · 2025-07-02

**Clarity:** 3
**Significance:** 2
**Originality:** 2
**Rating:** 5
**Confidence:** 4

**Summary:**

This paper proposes a learnable framework to enable LLMs' capability of learning when to think. It designs two types of thinking: short-form and long-form reasoning, and lets LLMs learn to choose the suitable reasoning type via the decoupled group relative policy optimization (DeGRPO) algorithm. The results on multiple Math benchmarks show that it can reduce 50%-90% thinking tokens.

**Questions:**

1. Why is it important to have the paired thinking response and the short response for the same problem in the warm-up? The data imbalance issue can also be avoided by sampling the same number of thinking and short responses from different sets of problems.  There could be some difficult problems that only have correct thinking responses. In this case, these hard problems will be removed from distillation.

2. What are the average lengths of the responses in think mode and short mode? How well is the model trained to follow the mode token to generate the corresponding response? For example, are the responses under the short mode always shorter than the thinking mode?

3. Do you think that if the RL is fully trained, the final model will finally learn to always generate shorter responses because of a higher reward? Because once the model learn to correctly solve the problem in the think mode, it can still be optimized to a shorter version.

**Ethical Concerns:**

["NO or VERY MINOR ethics concerns only"]

**Final Justification:**

The authors address my concerns with clarification and additional experiments.

**Limitations:**

They discuss the potential limitations in their paper, and there is no noticeable negative societal impact of their work.

**Quality:**

3

**Strengths And Weaknesses:**

Strengths:

* This paper focuses on the important problem in current long-chain-of-thought reasoning: the model may waste its thinking time on easy problems. It then proposes two thinking modes to make the reasoning more effective while keeping it accurate.
* To mitigate the imbalance between the single-mode token and the response, it modifies the original GRPO to decouple the mode loss and the generation loss. It introduces a hyperparameter to balance the optimization. Experimental results show that this hyperparameter can effectively avoid mode collapse caused by the length difference between the thinking mode and the response.
* It provides a comprehensive investigation on the training dynamics for thinking mode learning. It shows that LLMs will first learn to use long-chain outputs to improve accuracy, and then learn to generate more high-quality short answers to achieve a higher reward as defined in Section 3.2. The decoupling hyperparameter also has a high influence on the training stability.

Weaknesses:

* The idea is simple and straightforward: introducing the thinking mode tag to encourage the controlled generation under different models. The proposed DeGRPO is also an incremental improvement over the existing GRPO by adding a hyperparameter in special tokens (mode).
* The definition and boundary between the think mode and the short mode are not very clear.  Is there a specific length threshold between two modes? If these two modes are mainly distinguished by different distillation sources (general-purpose LLMs Qwen2.5-Math-1.5B-Instruct v.s. reasoning model DeepSeek-R1-Distill-Qwen-1.5B) , it may overfit on learning which model is more suitable to solve the problem, instead of learning the better thinking mode.
* More descriptions should be added for notations in Table 1. For example, the bold numbers are not the highest ones in the column. The baseline to calculate the token percentage is not mentioned.

---

> ### Author Rebuttal · Authors · 2025-07-30
>
> We sincerely appreciate the constructive comments and suggestions from Reviewer x1rq. We will revise our submission following the advice.
>
> ---
>
> >  **W1:** *The idea is simple and straightforward: introducing the thinking mode tag to encourage the controlled generation under different models. The proposed DeGRPO is also an incremental improvement over the existing GRPO by adding a hyperparameter in special tokens (mode).*
>
> **Response to W1**: Thank you for your valuable feedback. We acknowledge that our method may appear relatively simple and straightforward. Our primary goal is to explore what the minimally designed algorithm should look like for enabling automatic reasoning, such as the use of decoupled token to balance the learning of mode selection and response refinement.
>
> ---
>
> > **W2:** *(1) The definition and boundary between the think mode and the short mode are not very clear. (2) Is there a specific length threshold between two modes? (3) If these two modes are mainly distinguished by different distillation sources (general-purpose LLMs Qwen2.5-Math-1.5B-Instruct v.s. reasoning model DeepSeek-R1-Distill-Qwen-1.5B) , it may overfit on learning which model is more suitable to solve the problem, instead of learning the better thinking mode.*
>
> **Response to W2:** Thank you for the insightful comment.
> 1. We fully agree that the distinction between “long” and “short” is inherently relative, and there is no universal boundary. For example, a response length of 1,000 tokens may be considered long for GSM8K (average length = 624), yet relatively short for AIME (average length = 7,099).
> 2. Therefore, To address this variability, we do not rely on a fixed threshold to classify sequences as long or short. Instead, we adopt an implicit approach by leveraging the priors of two distinct LLMs, one optimized for reasoning and the other for non-reasoning tasks. This design  allows our method to adapt to tasks with diverse length distributions, avoiding the difficulty of selecting a explicit threshold.
> 3. The RL process involves two collaborative components: (1) fitting the mode and (2) generating a better response under the chosen mode. The first component aligns to the reviewer's comment, learning which model's knowledge is more suitable to solve the problem. However, the second component ensures that the model is not merely functioning as a "model classifier." Instead, it is jointly and adaptively optimized alongside the mode selection process.
>
> To further analyze this joint optimization, we categorize the samples in the Math-500 dataset into easy and hard problems based on the decisions made by our model. We then evaluate the performance of the base models (DeepSeek-R1-1.5B and Qwen-1.5B) on these subsets. As the reviewer suggested, the model indeed learns to distinguish problem difficulty. At the same time, the reinforcement learning model explores the potential of short responses across various problem. This results in a significant performance improvement on easy problems, increasing accuracy from 0.7531 to 0.9079.
>
> |  | Acc on Easy Problems   | Acc on Hard Problems |
> |:--| :--: | :--: |
> | R1-1.5B | 0.9163 (Think) | 0.5977 (Think) |
> | Qwen-1.5B | 0.7531 (Short) | 0.2835 (Short) |
> | Ours | 0.9079 (Short) | 0.6935 (Think) |
>
>
> ---
>
> > **W3:** *More descriptions should be added for notations in Table 1. For example, the bold numbers are not the highest ones in the column. The baseline to calculate the token percentage is not mentioned.*
>
> **Response to W3:** Thank you for the valuable suggestion. We will revise the descriptions accordingly by: (1) clearly highlighting the highest value within each method category (Base, Short CoT, and Hybrid); (2) clarifying that the percentage refers to the proportion of think-mode samples during evaluation, calculated as #Think / (#Think + #Short), where #Think and #Short denote the number of responses generated in think and short modes, respectively; and (3) incorporating additional details to enhance the clarity and overall readability of the table, as recommended.
>
> ---
>
> > **Q1:** *Why is it important to have the paired thinking response and the short response for the same problem in the warm-up? The data imbalance issue can also be avoided by sampling the same number of thinking and short responses from different sets of problems. There could be some difficult problems that only have correct thinking responses. In this case, these hard problems will be removed from distillation.*
>
> **Response to Q1:** Thank you for the insightful comment. We agree with the reviewer that paired data may not be strictly necessary. And data cleaning techniques could be employed to filter out low-quality samples. The reason why we use paired data is to ensure that the model is capable of generating both types of responses for all training samples. We believe this requirement can be relaxed in favor of improved data quality. We will include a discussion of this potential design in the future work section, as suggested by the reviewer.
>
> ---
>
> > **Q2:** What are the average lengths of the responses in think mode and short mode? How well is the model trained to follow the mode token to generate the corresponding response? For example, are the responses under the short mode always shorter than the thinking mode?
>
> **Response to Q2:** *Thank you for the insightful question! To address this, we explicitly guide the learned LLM to generate the desired responses by manually appending the control token. The following table presents length statistics over five runs on the validation dataset Math-500, demonstrating that the Think Mode produces longer responses.*
>
> | Mode | Length Mean | Length Std | | Acc |
> |:--:| :--: |:--:| :--: | :--:|
> | Think Mode | 3185 | 2446 | | 0.8436 |
> | Short Mode | 672 | 465 | | 0.7576 |
>
> We further compared the relative output lengths of the model given the same prompt. Across 2,500 trials on Math-500, we found that in 99.8% of cases, the output generated in Think Mode was longer than that in Short Mode. The remaining 0.2% (only 5 samples) of unexpected cases exhibited endless outputs, which may be due to the SFT phase, where certain examples were not perfectly covered by the training distribution.
>
> | Length Stats | len(Think)>len(Short) |
> |:--| :--: |
> | Warmup Model | 99.7% |
> | RL Model | 99.8% |
>
> ---
>
> > **Q3:** *Do you think that if the RL is fully trained, the final model will finally learn to always generate shorter responses because of a higher reward? Because once the model learn to correctly solve the problem in the think mode, it can still be optimized to a shorter version.*
>
> **Response to Q3:**  Thank you for your question. Shorter responses generally result in lower accuracy. In our reward setting, incorrect answers receive negative rewards and generating shorter responses tends to reduce the average reward. As a result, the training process will converges to a dynamic equilibrium, where the average rewards of both long and short responses become comparable. So, the final model will not always generate shorter responses.

---

> > ### Comment · Reviewer_x1rq · 2025-08-05
> >
> > Thanks for the detailed responses, which have solved my concerns. I will raise my score accordingly.

---

> > > ### Author Response · Authors · 2025-08-05
> > >
> > > We sincerely appreciate Reviewer x1rq's insightful suggestions and comments. We will polish the analysis regarding the distinction between long and short answers, as well as the related statistics, and incorporate them into the paper. In addition, following the reviewer’s advice, we will also include a discussion on the design choice of using paired data in SFT.
> > >
> > > Best regards,
> > > The Authors of Submission #2048

---

### Decision · Program_Chairs · 2025-09-17

**Decision:**

Accept (poster)

**Comment:**

This paper presents Thinkless, a learnable framework enabling LLMs to adaptively choose between short-form and long-form reasoning based on task complexity. The approach uses supervised fine-tuning with control tokens and a novel Decoupled Group Relative Policy Optimization (DeGRPO) algorithm that prevents mode collapse by separating optimization objectives for mode selection and response accuracy.

The paper addresses an important efficiency problem by tackling computational waste when models use extensive reasoning on simple problems. The technical solution is well-motivated, with DeGRPO effectively solving mode collapse through decoupled loss functions with clear theoretical justification. The authors provide comprehensive experimental analysis with thorough training dynamics evaluation across multiple mathematical benchmarks, demonstrating significant efficiency gains of 50-90% reduction in thinking tokens while maintaining comparable accuracy.

The distinction between short and long modes lacks clear definition and may overfit to distillation sources rather than true reasoning complexity. Additionally, the evaluation scope is limited primarily to mathematical reasoning, leaving generalizability to other domains uncertain.

Following the discussion phase, authors have adequately addressed major reviewer concerns regarding mode collapse solutions and experimental validation. The technical contribution of DeGRPO and practical importance of efficient reasoning allocation outweigh the limitations, making this a valuable contribution to the field. Thus I recommend to accept this paper.